**Subject Category:**
Biology (whole organism)

behaviour/ecology/health and disease and epidemiology

amphibian, horizontal transmission, *Janthinobacterium lividum*, *Lithobates clamitans*, tadpole

**Author for correspondence:**
Carl N. Keiser
e-mail: ckeiser@ufl.edu

# Tadpole body size and behaviour alter the social acquisition of a defensive bacterial symbiont

Carl N. Keiser[1], Trina Wantman[2], Eria A. Rebollar[3] and Reid N. Harris[4,5]

[1]Department of Biology, University of Florida, Gainesville, FL 32611, USA
[2]Department of Biological Sciences, University of Pittsburgh, Pittsburgh, PA 15260, USA
[3]Centro de Ciencias Genómicas, Universidad Nacional Autónoma de México, Cuernavaca, Mor, Mexico
[4]Amphibian Survival Alliance, London SW7 2HQ, UK
[5]Department of Biology, James Madison University, Harrisonburg, VA 22807, USA

CNK, 0000-0002-4936-7810

Individual differences in host phenotypes can generate heterogeneity in the acquisition and transmission of microbes. Although this has become a prominent factor of disease epidemiology, host phenotypic variation might similarly underlie the transmission of microbial symbionts that defend against pathogen infection. Here, we test whether host body size and behaviour influence the social acquisition of a skin bacterium, *Janthinobacterium lividum*, which in some hosts can confer protection against infection by *Batrachochytrium dendrobatidis*, the causative agent of the amphibian skin disease chytridiomycosis. We measured body size and boldness (time spent in an open field) of green frog tadpoles and haphazardly constructed groups of six individuals. In some groups, we exposed one individual in each group to *J. lividum* and, in other groups, we inoculated a patch of aquarium pebbles to *J. lividum*. After 24 h, we swabbed each individual to estimate the presence of *J. lividum* on their skin. On average, tadpoles acquired nearly four times more bacteria when housed with an exposed individual compared to those housed with a patch of inoculated substrate. When tadpoles were housed with an exposed group-mate, larger and 'bolder' individuals acquired more bacteria. These data suggest that phenotypically biased acquisition of defensive symbionts might generate biased patterns of mortality from the pathogens against which they protect.

## 1. Introduction

Global biodiversity loss is a chief concern among environmental disciplines, and identifying the mechanisms underlying this loss is

a major goal for ecologists. The threat of extinction is particularly alarming for amphibians, whose populations are declining worldwide [1–3]. One factor implicated in the unprecedented decline of amphibians is the fungal pathogen *Batrachochytrium dendrobatidis* (Bd), the causative agent of the skin disease chytridiomycosis [4]. Bd has been detected in over 500 amphibian species and outbreaks have occurred in over 50 countries [5], some of which have led to the loss of over 40% of local amphibian diversity [6,7]. One proposed reason that some species are resistant to Bd is that they harbour skin bacteria that defend against infection. Indeed, a proposed mitigation strategy is bioaugmentation, where skin bacteria that confer protection against Bd are prophylactically applied to susceptible hosts [8–12]. Unfortunately, this individual-level treatment is unlikely to be a realistic population-wide management strategy [8]. Thus, investigating the degree to which experimentally exposed individuals can serve as sources for further transmission of defensive bacteria among hosts will inform us as to the efficacy of this strategy.

Hosts' rapid acquisition of defensive symbionts can be an important safeguard when populations are confronted with novel pathogens [13]. To be most impactful for bioaugmentation, a defensive symbiont should therefore be transmissible across susceptible hosts via direct interaction or indirectly via the environment [11]. The bacterium *Janthinobacterium lividum* releases antifungal metabolites which can protect hosts against Bd [14], and it has been found in the skin microbiomes of several amphibian families [15]. The prophylactic application of *J. lividum* within and among amphibian species has shown dramatic reductions in chytridiomycosis mortality (100% reduction in some cases, but totally ineffective in others; [16]) [10,14,17]. Furthermore, *J. lividum* can be horizontally transmitted between green frog (*Lithobates clamitans*) tadpoles via direct physical contact and indirectly through shared substrates [18]. Some tadpoles form dense aggregations potentially to reduce predation risk or facilitate foraging [19], and some tadpoles prefer to be near conspecifics [20]. Tadpoles can acquire Bd infection on their keratinized mouthparts ('oral chytridiomycosis'; [21,22,23]), potentially through aggregating with infected conspecifics or via the environment [24]. Tadpole aggregation may also contribute to the acquisition of defensive symbionts [25], potentially safeguarding individuals until adulthood [18]. However, individual differences in morphological and behavioural traits may underlie variation among individuals in bacterial acquisition.

Previous studies on tadpoles have found temporally consistent differences among individuals in the proportion of time they spend active [26], their risk-taking behaviour, and the degree to which they explore novel environments [27]. Differences in tadpole behaviour can influence parasite infection risk: for example, more inactive or less exploratory wood frog tadpoles have greater trematode infection intensity [28], and wood frog tadpoles that take longer to find food have greater ranavirus infection loads [29]. These types of behavioural measures may similarly predict the acquisition of symbiotic microbes that defend against parasite infection, but have not yet been tested. Here, we use green frog tadpoles to ask the following questions: (1) Do individuals acquire more bacteria from the environment or from a group-mate? (2) To what degree do tadpole behaviour and body size generate differences in the social acquisition of defensive bacteria?

# 2. Material and methods

## 2.1. Amphibian collection

We collected five *L. clamitans* egg masses in May 2017 between 09.00 h and 11.00 h in fish hatchery ponds at the Linesville State Fish Hatchery in Linesville PA and transported them to 150 l plastic wading pools at the Donald S. Wood Field Laboratory. Green frog tadpoles exhibit intermediate levels of aggregation compared to other anurans [30], making them a valuable subject to test for social versus environmental acquisition of microbes. As tadpoles emerged from egg masses, they were transferred to 500 l outdoor mesocosms initiated with 200 g dried leaves, 15 g rabbit chow and 2 l of lake water and covered with shade cloth. Each mesocosm contained 110 tadpoles and was maintained for 90 days under natural conditions. After growing to at least Gosner stage 26, a subset of the tadpoles was transported into the laboratory and maintained in 19 groups of six siblings for the remainder of the experiment. The remaining tadpoles were returned to their pond of origin. We gave each individual a unique subdermal ID tag using visible implant elastomer tags (VIE tags; Northwestern Marine Technology Inc. Shaw Island, WA, USA) following methods in [31], and tadpoles were provided ad libitum food for 72 h as part of an acclimation period. We then measured each tadpole's head length using digital calipers.

## 2.2. Experimental overview

We measured body size and behaviours of green frog tadpoles that were maintained in 19 social groups each containing six individuals. In the *Social Acquisition* experiment (*n* = 10 groups), we isolated a haphazardly

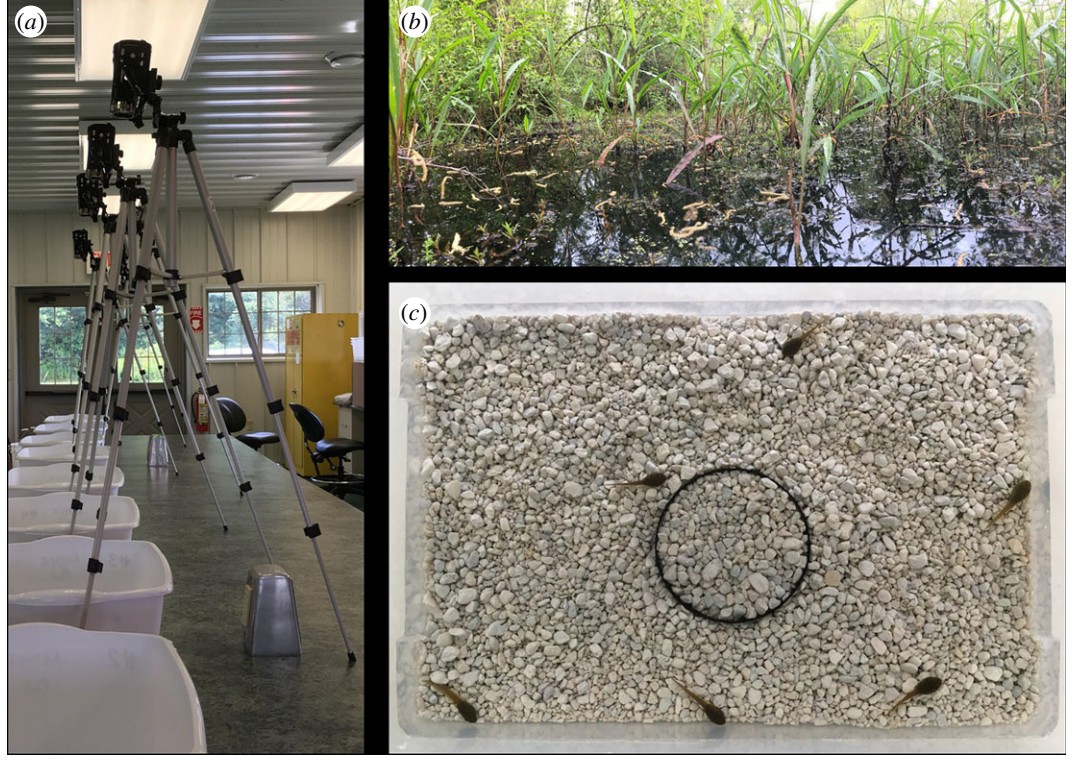

**Figure 1.** Experimental set-up. (*a*) Tripod-mounted cameras used to film behavioural assays. (*b*) Pond-edge habitat from which egg masses were collected. Tadpoles were placed individually into 15 l tubs and filmed for 10 min to quantify the proportion of time spent at the edge of the tub versus the centre. (*c*) Image of environmental exposure experiment. Tadpoles spent 24 h in a 15 l tub containing a white pebble substrate with a 100 mm diameter section containing pebbles that had prior exposure to *J. lividum*. The exposed section is denoted with black edges.

selected individual and exposed it to the defensive bacterial symbiont *J. lividum* for 24 h. We then placed the exposed individuals and their unexposed group-mates into a separate tub and allowed them to interact naturally for 24 h. In the *Environmental Acquisition* experiment (*n* = 9 groups), we exposed a patch of aquarium pebbles to *J. lividum* and then placed it into a field of unexposed, sterilized pebbles. We then moved social groups from their housing tubs into the tubs containing exposed pebble patches and left them undisturbed for 24 h (figure 1*c*). Finally, we isolated each individual and swabbed their skin to estimate the presence of *J. lividum* by counting bacterial colony-forming units on selective growth media.

## 2.3. Behavioural assays

Tadpoles were moved from their groups into 15 l clear tubs filled with 5 l of aged well-water. Each tub sat atop a 10 cm$^2$ grid paper, and the edges were encompassed by an unmarked 2 cm layer. Individuals were kept under a black metal mesh cup in the centre of the tub for a 10 min acclimation period after which the cup was removed. We used a Besteker Portable Camcorder stationed above the tub to record each tadpole's behaviour for 10 min and later quantified the proportion of time each individual spent at the tub edge versus in the open centre (figure 1). This measure is an estimate of 'boldness', where individuals that spend less time at the safer edge of the container and more time in the riskier open field are considered bolder [27,32]. Individuals were tested twice in two consecutive days, and the value from the two assays was averaged to represent an individual's boldness.

## 2.4. Bacterial acquisition experiments

We grew a rifampin-resistant *J. lividum* strain for this experiment that could be collected and distinguished from non-rifampin-resistant bacteria naturally present on tadpole skin, following methods in [18]. We re-plated and maintained the rifampin-resistant bacteria on 1% rifampin tryptone agar plates. Then, we grew a liquid bacterial culture by picking a single bacterial colony from a 1% rifampin tryptone agar plate and growing in 1% rifampin tryptone until it reached a concentration of

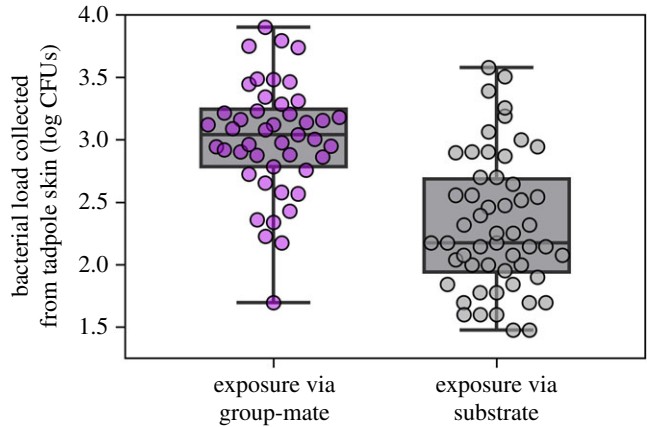

**Figure 2.** Acquisition of *J. lividum* bacteria across two exposure regimes. When tadpoles are exposed to *J. lividum* via a group-mate, individuals acquire more bacteria compared to those exposed via a patch of pebble substrate. The boxes extend from the 25th to 75th percentiles, the horizontal line represents the median and the vertical lines extend to the minimum and maximum values.

approximately $1.7 \times 10^7$ colony-forming units/ml. For *Social Acquisition*, a single individual was selected haphazardly from each group and exposed to *J. lividum* by isolating them in 500 ml cups filled with 200 ml sterile aged well water and 50 ml of liquid bacterial broth. The tadpoles were maintained in the same laminar-flow hood as the Petri dishes. Then, exposed individuals were rinsed under aged well-water for 5 s and added with the rest of their original social group to a 15 l tub filled with 5 l of aged well-water for 24 h. For *Environmental Acquisition*, a 50 ml Petri dish was filled with autoclave-sterilized aquarium stones and 50 ml of liquid *J. lividum* broth was added to the dish, covered and maintained for 24 h in a laminar-flow hood at room temperature (approx. 21°C) under natural light schedule. The Petri dishes were rinsed with aged well-water for 5 s to remove the liquid bacterial culture and then set inside a field of sterilized aquarium pebbles in a 15 l tub filled with 5 l of aged well-water. After 2 h, a group of six tadpoles were added to the container edges and allowed to move around the tub naturally for 24 h.

The following day, we collected each individual from both exposure regimes (Environmental and Social) and swabbed their skin back and forth five times on both the lateral sides, dorsal and ventral sides, and on the mouth (following techniques in [18]). Swabs were placed directly into 1.5 ml Eppendorf tubes of phosphate-buffered saline, vortexed for 5 min, and used to make a 3-step serial dilution. We plated 100 µl of each dilution on 1% rifampin tryptone plates and incubated at ambient temperatures in a laminar flow hood. We estimated the concentration of bacteria on tadpoles' skin by counting the number of bacterial colonies on the plates after 48 h (colony-forming units, CFUs). We maintained two control groups that had never been exposed to our rifampin-resistant *J. lividum* strain, one with autoclaved pebbles in the bottom and one with a tadpole which had spent 24 h in 200 ml of water without *J. lividum*. We swabbed these 12 individuals, and the pebbles, and found no detectable amount of *J. lividum*.

## 2.5. Statistical analyses

All CFU values were log-transformed and normality of model residuals was confirmed. We used a general linear mixed model (GLMM) to test whether tadpoles acquired more bacteria from social acquisition or environmental acquisition. We used a GLMM to test for individual bacterial load in the social acquisition experiment with head length, boldness (proportion of time spent in open field), and the CFUs collected from the experimentally exposed individual value as predictor variables. The interaction term between tadpole body size and boldness was not significant and was removed for model simplification [33]. Individual ID nested in experimental group ID and source clutch ID was included as a random effect in both models.

## 3. Results and discussion

Based on our estimates from skin swabs, we found that tadpoles acquired nearly four times as many *J. lividum* bacteria in the presence of an exposed group-mate compared to tadpoles that were housed with a patch of exposed substrate ($F_{1,17} = 29.76$, $p < 0.0001$; figure 2). It may be that individuals simply

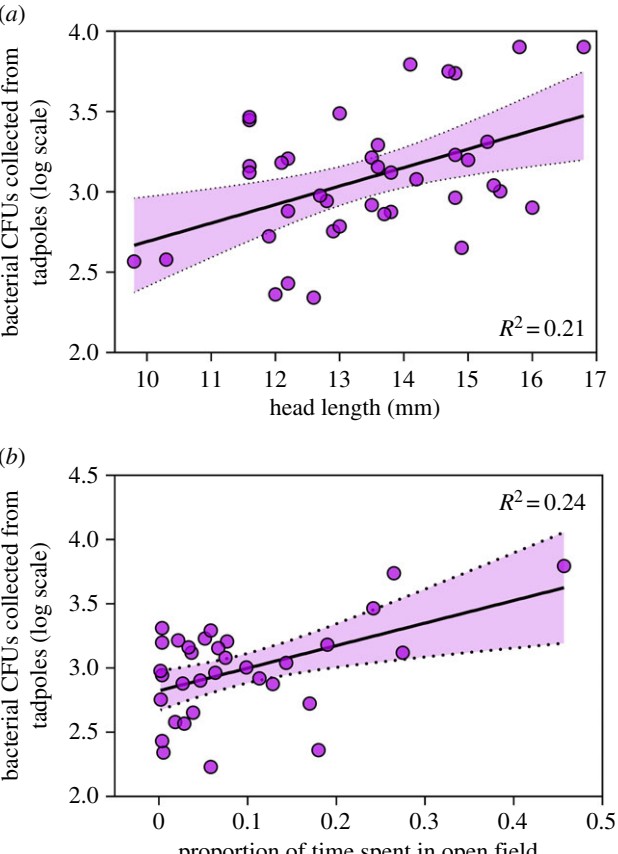

**Figure 3.** Phenotypic predictors of bacterial acquisition. Tadpoles acquired more bacteria if they were (*a*) larger and (*b*) bolder, meaning they spent more time in the open compared to alongside the container edge.

interacted less with the exposed patch than they did with an exposed group-mate, though Rebollar *et al.* [18] found that an environmental reservoir for indirect transmission of *J. lividum* can increase bacterial acquisition relative to direct transmission in tadpole pairs. A potential means of increasing environmental acquisition could be to simultaneously inoculate the substrate with *J. lividum* and periphyton to attract tadpoles to the exposed area. Although environmental acquisition of cutaneous microbiota is clearly important in amphibians [34,35], these data suggest that social interactions in a group setting can potentially increase the spread of defensive bacteria among hosts. A previous experiment showed that *L. clamitans* aggregation behaviour was not influenced by tadpole density [30], though future studies should manipulate host density to test the degree to which conspecific proximity may influence transmission dynamics across different social contexts.

When tadpoles were housed with an exposed group-mate, we also found that larger individuals acquired more *J. lividum* bacteria on their skin ($F_{1,28} = 6.59$, $p = 0.02$; figure 3*a*). Our swabbing protocol was standardized among individuals, so it is unlikely that this pattern is an artefact of larger individuals simply being swabbed more. So, larger individuals may simply have more body surface area on which to acquire microbes. Larger individuals may engage in more social interactions within groups or simply move around the environment to a greater degree [36]. Further, larger tadpoles are often more aggressive towards smaller tadpoles [37,38], so antagonistic interactions may play a role in transmission dynamics. We also found that bolder individuals (those that spent more time in the open during open-field trials) acquired more bacteria ($F_{1,17} = 12.21$, $p = 0.002$; figure 3*b*). This trend matches what previous studies have found with bolder tadpoles experiencing increased infection intensity with parasites [28,29]. Although a trend between host personality and Bd infection has not yet been identified at any amphibian life stage, if the same traits underlie the acquisition of both defensive bacteria and the pathogens against which they confer protection, these highly 'competent' individuals may play a crucial role in disease dynamics at the population level [39,40]. Further, we expect that the proportion of time that tadpoles spend in the open (i.e. 'boldness'), or other space-use traits, will generate interesting trade-offs among predation risk, pathogen exposure and symbiont acquisition that warrant future investigation.

It is unlikely that exposure to *J. lividum* at the tadpole stage can safeguard individuals against Bd infection after metamorphosis [41]. However, this protection may help individuals reach adulthood, or at least reach a point where the host's adaptive immune system matures and becomes more effective against the pathogen [11]. Further, tadpoles that are more active and exploratory may maintain these behavioural phenotypes across metamorphosis [26], so future studies should test whether the traits that predict individual bacterial acquisition in larval anurans similarly predict bacterial acquisition in adults. For chytridiomycosis, behaviour may be particularly important in some cases where innate immune mechanisms appear to be insufficient in defense against Bd [42]. These data suggest that phenotypically biased acquisition of defensive symbionts might generate similarly biased patterns of mortality from the pathogens against which they protect. Indeed, we found that larger tadpoles acquired more defensive bacteria, while Valencia-Aguilar et al. [24] found that larger tadpoles are more likely to be Bd positive. From a conservation perspective [43,44], perhaps the inoculation of highly competent transmitters would increase the spread of defensive symbionts, potentially augmenting group or population resilience against Bd compared to the inoculation of inferior transmitters.

Ethics. All procedures were conducted under the supervision and approval of IACUC (# 17060844) at the University of Pittsburgh.

Data accessibility. The raw data associated with this manuscript are available at Dryad DigitalRepository: https://doi.org/10.5061/dryad.mn8v722 [45].

Authors' contributions. C.N.K., E.A.R and R.N.H. designed the study; C.N.K. and T.W. scored the behavioural videos, and C.N.K. carried out the experiments, analysed the data and prepared the manuscript. All authors gave final approval for publication and agree to be accountable for all aspects of the work.

Competing interests. The authors declare that we that we have no competing interests.

Funding. This study was funded by a Frank J. Schwartz Early Career Research Fellowship and the G. Murry McKinley Research Fund at the Pymatuning Laboratory of Ecology, and the Rice University Academy of Fellows.

Acknowledgements. We thank Laura Brannelly, Michel Ohmer and Veronica Saenz for advice on field collection and experimental design and Brandon LaBumbard and Doug Woodhams for supplying bacterial stocks. We also thank the staff at the Pymatuning Lab of Ecology.

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
