## [Reviewer comments · Royal Society Open Science]

Review History

RSOS-191080.R0 (Original submission)

Review form: Reviewer 1

Is the manuscript scientifically sound in its present form?

Yes

Are the interpretations and conclusions justified by the results?

Yes

Is the language acceptable?

Yes

Do you have any ethical concerns with this paper?

No

Have you any concerns about statistical analyses in this paper?

No

Recommendation?

Accept with minor revision (please list in comments)

Comments to the Author(s)

Dear,

Dr John Dalton, Associate Editor of Royal Society Open Science journal

Here I sending my review on the Manuscript ID RSOS-191080, entitled "Tadpole body size and behavior alter the social acquisition of a defensive bacterial symbiont."

The authors (Keiser et al.) details an interesting work that tested the effect of social interaction in the transmission of a microbial symbiont in the green frog tadpoles. In some host this bacterium can confer protection against infection by *Batrachochytrium* a lethal pathogen that cause high level of mortality in amphibians. Clearly the results present by the authors have merits to be publish. However, I think the manuscript could be improved after some revisions that I details below

Sincerely yours

Is not clear in the introduction if the tadpoles of green frog are social or sub-social, i.e. they form schools or only aggregate for feeding or because they select warm sites as another species (e.g. toad tadpoles). The methodology need an explicative figure for the device use to in the behavioral assay. In the current form is difficult to follow all the steep for each experiment.

The authors used only one density of tadpoles to study the transmission of the bacterium, so how do they know the effect of tadpole aggregation in the transmission process. In the other hand sometimes large tadpoles shown aggressive behavior over the smaller tadpoles, the authors observed this pattern in the experiments?

Tadpoles exposed to a patch of pebbles previously colonized by the bacterium may have no attractive for tadpoles, perhaps if the patch were colonized also by periphyton the density of acquire bacteria could be similar to the group-mate treatment

For me is not surprise that large tadpole acquire more bacteria than smaller one, because the large tadpoles have major expose area, then they have more area available for colonized by bacteria.

Line 188 past metamorphosis, replace by after metamorphosis, and also this paragraphs is little speculative

Line 190-193 a little speculative

Line 182-183 similarly is expected that more active tadpole is more exposed to visual predators, therefore the frequency of encounter with predator, parasites and bacteria is probably major in more active individuals. Here may be exist a new trade-off between acquired bacteria for protection against pathogens and exposed to visual predators. Will be interesting study how interact these variables with the presence /absence of predators.

Decision letter (RSOS-191080.R0)

08-Jul-2019

Dear Dr Keiser

On behalf of the Editors, I am pleased to inform you that your Manuscript RSOS-191080 entitled "Tadpole body size and behavior alter the social acquisition of a defensive bacterial symbiont" has been accepted for publication in Royal Society Open Science subject to minor revision in

accordance with the referee suggestions. Please find the referees' comments at the end of this email.

The reviewers and handling editors have recommended publication, but also suggest some minor revisions to your manuscript. Therefore, I invite you to respond to the comments and revise your manuscript.

- Ethics statement

- Data accessibility

If you wish to submit your supporting data or code to Dryad (<http://datadryad.org/>), or modify your current submission to dryad, please use the following link:
<http://datadryad.org/submit?journalID=RSOS&manu=RSOS-191080>

- Competing interests

- Authors' contributions

- Acknowledgements

- Funding statement

Because the schedule for publication is very tight, it is a condition of publication that you submit the revised version of your manuscript before 17-Jul-2019. Please note that the revision deadline will expire at 00.00am on this date. If you do not think you will be able to meet this date please let me know immediately.

Supplementary files will be published alongside the paper on the journal website and posted on the online figshare repository (<https://rs.figshare.com/>). The heading and legend provided for each supplementary file during the submission process will be used to create the figshare page,

so please ensure these are accurate and informative so that your files can be found in searches. Files on figshare will be made available approximately one week before the accompanying article so that the supplementary material can be attributed a unique DOI.

on behalf of Dr John Dalton (Associate Editor) and Kevin Padian (Subject Editor)
openscience@royalsociety.org

Associate Editor Comments to Author (Dr John Dalton):

Dear Authors,

your paper has received a favourable review but please address minor comments from reviewer.

Reviewer comments to Author:
Reviewer: 1

Comments to the Author(s)
Royal Society Open Science
Dear,

Dr John Dalton, Associate Editor of Royal Society Open Science journal
Here I am sending my review on the Manuscript ID RSOS-191080, entitled "Tadpole body size and behavior alter the social acquisition of a defensive bacterial symbiont."

The authors (Keiser et al.) details an interesting work that tested the effect of social interaction in the transmission of a microbial symbiont in the green frog tadpoles. In some host this bacterium can confer protection against infection by *Batrachochytrium* a lethal pathogen that cause high level of mortality in amphibians. Clearly the results present by the authors have merits to be published. However, I think the manuscript could be improved after some revisions that I details below

Sincerely yours

Is not clear in the introduction if the tadpoles of green frog are social or sub-social, i.e. they form schools or only aggregate for feeding or because they select warm sites as another species (e.g. toad tadpoles). The methodology need an explicative figure for the device use to in the behavioral assay. In the current form is difficult to follow all the steep for each experiment.

The authors used only one density of tadpoles to study the transmission of the bacterium, so how do they know the effect of tadpole aggregation in the transmission process. In the other hand sometimes large tadpoles shown aggressive behavior over the smaller tadpoles, the authors observed this pattern in the experiments?

Tadpoles exposed to a patch of pebbles previously colonized by the bacterium may have no attractive for tadpoles, perhaps if the patch were colonized also by periphyton the density of acquire bacteria could be similar to the group-mate treatment

For me is not surprise that large tadpole acquire more bacteria than smaller one, because the large tadpoles have major expose area, then they have more area available for colonized by bacteria. Line 188 past metamorphosis, replace by after metamorphosis, and also this paragraphs is little speculative

Line 190-193 a little speculative

Line 182-183 similarly is expected that more active tadpole is more exposed to visual predators, therefore the frequency of encounter with predator, parasites and bacteria is probably major in more active individuals. Here may be exist a new trade-off between acquired bacteria for protection against pathogens and exposed to visual predators. Will be interesting study how interact these variables with the presence /absence of predators.

Author's Response to Decision Letter for (RSOS-191080.R0)

See Appendix A.

Decision letter (RSOS-191080.R1)

01-Aug-2019

Dear Dr Keiser,

I am pleased to inform you that your manuscript entitled "Tadpole body size and behavior alter the social acquisition of a defensive bacterial symbiont" is now accepted for publication in Royal Society Open Science.

Royal Society Open Science operates under a continuous publication model (<http://bit.ly/cpFAQ>). Your article will be published straight into the next open issue and this will be the final version of the paper. As such, it can be cited immediately by other researchers.

As the issue version of your paper will be the only version to be published I would advise you to check your proofs thoroughly as changes cannot be made once the paper is published.

on behalf of Dr John Dalton (Associate Editor) and Kevin Padian (Subject Editor)
openscience@royalsociety.org

Appendix A

Dear Dr. Power and Dr. Dalton,

My collaborators and I are pleased to submit a revised version of our manuscript titled "**Tadpole body size and behavior alter the social acquisition of a defensive bacterial symbiont**" (RSOS-191080) for reconsideration as a research paper in *Royal Society Open Science*. We are confident this manuscript has been significantly improved by the fastidious examination of the reviewer and editor.

The reviewer made several interesting points about our experimental design and interpretation of data, all of which have been added to our revised manuscript.. One major change is that we have added a new figure, as suggested by the reviewer, to depict our experimental design. We address the reviewer's comments point-by-point below and have highlighted our revisions in the manuscript body.

We thank the editors and reviewers for their patience, time, and efforts, and we look forward to future interactions with *Royal Society Open Science*.

Sincerely,

Carl Nick Keiser

Department of Biology
University of Florida
Email: ckeiser@ufl.edu

Referee: 1

1. Is not clear in the introduction if the tadpoles of green frog are social or sub-social, i.e. they form schools or only aggregate for feeding or because they select warm sites as another species (e.g. toad tadpoles).

Author Response: Thanks for pointing this out. Green frog tadpoles are not described as “social” or “sub-social” in the literature (at least not that I have found). However, they are often found in aggregations at pond-edges. We have now added more details to the introduction as to why tadpoles aggregate (lines 68-71): “Some tadpoles form dense aggregations potentially to reduce predation risk or facilitate foraging [19], and some tadpoles prefer to be near conspecifics [20]. Tadpole aggregation may also contribute to the acquisition of defensive symbionts [21], potentially safeguarding individuals until adulthood [18].” And added a citation to the Methods section in reference to green frog tadpole aggregation: “Green frog tadpoles exhibit intermediate levels of aggregation compared to other anurans [26], making them a valuable subject to test for social vs. environmental acquisition of microbes.” (lines 89-91).

2. The methodology need an explicative figure for the device use to in the behavioral assay. In the current form is difficult to follow all the step for each experiment.

Author Response: Are you referring to the behavioral observations? We can upload a photograph of the setup, it is up to the editor to decide if it should be a new figure or a supplemental document. We have treated it like a new figure in the revised version (Figure 1), but are happy to change depending on the editors’ thoughts. We have included a new figure caption to help explain the experimental setup: “**Figure 1.** Experimental setup. (top) Pond-edge habitat from which egg masses were collected. (left) Tripod-mounted cameras used to film behavioral assays. Tadpoles were placed individually into 15L tubs and filmed for 10 minutes to quantify the proportion of time spent at the edge of the tub vs. the center. (bottom) Image of environmental exposure experiment. Tadpoles spent 24hr in a 15L tub containing a white pebble substrate with a 100mm diameter section containing pebbles that had prior exposure to *J. lividum*. The exposed section is denoted with black edges.” which we now reference throughout the methods section. I hope this helps clarify our methods.

3. The authors used only one density of tadpoles to study the transmission of the bacterium, so how do they know the effect of tadpole aggregation in the transmission process.

Author Response: This is a great question, and something we should have added to the manuscript. A previous study showed that tadpole density does not alter aggregation behavior in green frog tadpoles (Smith, G.R. & Jennings, 2004 Spacing of the Tadpoles of *Hyla versicolor* and *Rana clamitans*), but it would be interesting to see if these transmission dynamics are density-dependent. We have added this to the discussion section: “Although a previous experiment showed that *L. clamitans* aggregation behavior was not influenced by tadpole density [26], future studies should manipulate host density to test the degree to which conspecific proximity may influence transmission dynamics across different social contexts.” (lines 197-182).

4. In the other hand sometimes large tadpoles shown aggressive behavior over the smaller tadpoles, the authors observed this pattern in the experiments?

Author Response: We did not observe any agonistic interaction, but this would be interesting. We have added this as a potential transmission mode in our discussion: “Larger individuals may engage in more social interactions within groups or simply move around the environment to a greater degree [32]. Further, larger tadpoles are often more aggressive towards smaller tadpoles [33, 34], so antagonistic interactions may play a role in transmission dynamics.” (lines 187-190).

5. Tadpoles exposed to a patch of pebbles previously colonized by the bacterium may have no attractive for tadpoles, perhaps if the patch were colonized also by periphyton the density of acquire bacteria could be similar to the group-mate treatment

Author Response: Another great point! We weren't trying to attract the tadpoles to the substrate, but rather wanted natural activity patterns to influence transmission. We have added a statement as you suggested though: “A potential means of increasing environmental acquisition could be to simultaneously inoculate the substrate with *J. lividum* and periphyton to attract tadpoles to the exposed area.” (lines 175-177).

6. For me is not surprise that large tadpole acquire more bacteria than smaller one, because the large tadpoles have major expose area, then they have more area available for colonized by bacteria.

Author Response: Yes, we agree with this interpretation, as stated in the discussion: “Our swabbing protocol was standardized among individuals, so it is unlikely that this pattern is an artifact of larger individuals simply being swabbed more. So, larger individuals may simply have more body surface area on which to acquire microbes.” (lines 186-187).

7. Line 188 past metamorphosis, replace by after metamorphosis, and also this paragraphs is little speculative

Author Response: We have changed the wording as suggested, and we agree that this paragraph is speculative. As far as we know, this is the first experiment of its type so we are forced to be a bit speculative. I think these types of ideas proposed in this paragraph are important for inspiring future studies. We have tried to dwindle the speculative nature of the paragraph, however, or at least demonstrate that these are speculations.

8. Line 190-193 a little speculative

Author Response: Yes, this is speculative, but all of the statements are supported by citations, and I think these types of statements are what inspire future experiments, so I think they are important to keep. We have changed out wording to be more clear that these are speculations.

9. Line 182-183 similarly is expected that more active tadpole is more exposed to visual predators, therefore the frequency of encounter with predator, parasites and bacteria is probably major in more active individuals. Here may be exist a new trade-off between acquired bacteria for protection against pathogens and exposed to visual predators. Will be interesting study how interact these variables with the presence /absence of predators.

Author Response: Great point! We have added a statement like this to our discussion section: “Further, we expect that the proportion of time that tadpoles spend in the open (i.e., “boldness”), or other space-use traits, will generate interesting tradeoffs between predation

risk, pathogen exposure, and symbiont acquisition that warrant future investigation.” (lines 197-200).